# Rate of brain aging and *APOE ε4* are synergistic risk factors for Alzheimer's disease

Christin A Glorioso[1,2,*] , Andreas R Pfenning[3,*], Sam S Lee[1,2] , David A Bennett[4], Etienne L Sibille[5,6], Manolis Kellis[7,8,†], Leonard P Guarente[1,2,9,†]

Advanced age and the *APOE ε4* allele are the two biggest risk factors for Alzheimer's disease (AD) and declining cognitive function. We describe a universal gauge to measure molecular brain age using transcriptome analysis of four human postmortem cohorts (n = 673, ages 25–97) free of neurological disease. In a fifth cohort of older subjects with or without neurological disease (n = 438, ages 67–108), we show that subjects with brains deviating in the older direction from what would be expected based on chronological age show an increase in AD, Parkinson's disease, and cognitive decline. Strikingly, a younger molecular age (−5 yr than chronological age) protects against AD even in the presence of *APOE ε4*. An established DNA methylation gauge for age correlates well with the transcriptome gauge for determination of molecular age and assigning deviations from the expected. Our results suggest that rapid brain aging and *APOE ε4* are synergistic risk factors, and interventions that slow aging may substantially reduce risk of neurological disease and decline even in the presence of *APOE ε4*.

## Introduction

Quality of life in old age is often compromised by dementia, mild cognitive impairment, and declining mobility. Dementia is common in the elderly with a prevalence that rises with age from about 5% in people 71–79 yr old to nearly 40% in people older than 90 yr in the United States (Plassman et al, 2007). Alzheimer's disease (AD) is the most common form of dementia (about 70% of cases), with vascular dementia, Lewy body dementia, frontotemporal dementia, and Parkinson's disease (PD) making up the majority of other cases (Plassman et al, 2007). Many people (20–30%) meet criteria for more

than one type of dementia (mixed dementia) (Jellinger, 2013). AD is characterized clinically by progressive memory impairment, declining judgment, and increased mood symptoms, leading to eventual loss of most cognitive function and death. Pathological features of AD include irreversible neuronal loss, particularly in the hippocampus and temporal cortex, extracellular *β* amyloid plaques and neurofibrillary tangles (Nussbaum & Ellis, 2003). The overwhelming majority of AD cases are late onset (LOAD) (~93%) and nonfamilial (99%) (Nussbaum & Ellis, 2003). The two biggest risk factors for LOAD are advanced age and the presence of *ε4* alleles of the *APOE* gene. How these risk factors relate to each other is an open question.

Studies have characterized age-related differences in transcription in the human brain (Lu et al, 2004; Erraji-Benchekroun et al, 2005; Glorioso et al, 2011). These studies show decreases in expression of neuronal synaptic-related genes, calcium signaling, and DNA damage–related genes and increases in glial inflammation–related genes with age (Yankner et al, 2008; Glorioso & Sibille, 2011). These transcriptional changes have been used as a gauge to assign a molecular age to any brain sample and can identify brains that deviate from the expected based on chronological age; that is, brains showing unusually slow or fast aging compared with the average. A few studies have further interrogated the intersection of normal brain aging and AD and have generally supported an overlap between normal age-related transcriptional differences in the brain and differences between AD and control subjects (Miller et al, 2008; Cao et al, 2010; Avramopoulos et al, 2011; Saetre et al, 2011). Two studies showed aging acceleration in AD subjects versus controls (Cao et al, 2010; Saetre et al, 2011). These studies are based on small cohorts (n < 50 subjects) (Miller et al, 2008; Cao et al, 2010; Avramopoulos et al, 2011; Saetre et al, 2011) and do not relate brain aging to known genetic risk factors for AD.

Another gauge that has proved robust to assign molecular age is the DNA methylation clock. Quantitative assessment of many

---

[1]Department of Biology, Massachusetts Institute of Technology, Cambridge, MA, USA  [2]Paul F. Glenn Center for Biology of Aging Research at Massachusetts Institute of Technology, Cambridge, MA, USA  [3]Department of Computational Biology, School of Computer Science, Carnegie Mellon University, Pittsburgh, PA, USA  [4]Rush Alzheimer's Disease Center, Rush University Medical Center, Chicago, IL, USA  [5]Department of Psychiatry and of Pharmacology and Toxicology, University of Toronto, Toronto, Ontario, Canada  [6]Campbell Family Mental Health Research Institute, Toronto, Ontario, Canada  [7]Computer Science and Artificial Intelligence Laboratory, Massachusetts Institute of Technology, Cambridge, MA, USA  [8]The Broad Institute of Harvard and the Massachusetts Institute of Technology, Cambridge, MA, USA  [9]The Koch Institute, Massachusetts Institute of Technology, Cambridge, MA, USA

Correspondence: leng@mit.edu; glorioso@mit.edu
*Christin A Glorioso and Andreas R Pfenning contributed equally to this work
†Manolis Kellis and Leonard P Guarente contributed equally to this work

---

 

5'—C—phosphate—G—3' (CpG) methylation sites has defined a subset of 300–500 sites whose level of methylation correlates very strongly with chronological age (Horvath, 2013), and this epigenetic age has been associated with AD (Levine et al, 2015). An important difference between transcriptomic age and methylation age is that the latter uses the same methylation sites to assign age to many different tissue types in humans (Horvath, 2013).

Besides advanced age, the biggest risk factor for LOAD is an allelic variant of the *APOE* gene. More than 15 genome-wide association studies have implicated *APOE ε4* in AD, making it by far the most consistent genetic risk factor (Bertram et al, 2010; Lambert et al, 2013). *APOE* encodes apolipoprotein E, a constituent of the low-density lipoprotein particle involved in clearance of cholesterol and a component of amyloid plaques (Nussbaum et al, 2015). There are three human variants of the gene: *APOE ε2* (cys112, cys158), *APOE ε3* (cys112, arg158), and *APOE ε4* (arg112, arg158) (Nussbaum et al, 2015). *APOE ε3* is considered the wild-type allele and is the most common genotype with an allele frequency about 76% (Warren J. Strittmatter, 1996). *APOE ε4*, with an allele frequency of about 14%, increases the lifetime risk of AD by twofold to fourfold (Nussbaum et al, 2015; Nussbaum and Ellis, 2003). There also appears to be a dose effect, in that disease-free survival was shown to be lower in homozygotes compared with heterozygous. Consistent with these findings, *APOE ε4* alleles shift the age at onset earlier in the presence of one allele and earlier still in the presence of two alleles (Nussbaum et al, 2015). *APOE ε4* has also been shown to be a risk factor for rate of age-related cognitive decline, even without AD-associated pathology (Glorioso & Sibille, 2011). *APOE ε4* may also be a risk factor for other dementias, including dementia with Lewy bodies (Tsuang et al, 2013; Bras et al, 2014) and perhaps PD, although the results of these studies are mixed (Williams-Gray et al, 2009) with some showing significant risk and others showing none.

Here, we develop a reliable transcriptome-based gauge of molecular brain aging and use this tool to determine how rapidly brains have aged compared with the average in a cohort. This method correlates well with the DNA methylation-based clock (Horvath, 2013). We use the inferred aging rate to assess whether molecular brain aging is a risk factor for AD and a variety of other late life maladies in a large naturalistic cohort of older subjects. Moreover, we relate brain aging and *APOE ε4* status as risk factors for AD, focusing most heavily on LOAD and cognitive aging, for which we are most powered. Our findings suggest that brain aging is an important risk factor for AD and acts synergistically with *APOE ε4*

and may have important therapeutic implications for treating this and other late onset brain diseases.

## Results

### Overview

We used five human postmortem brain cohorts to develop and test a transcriptome-based biological brain age gauge. Characteristics of these cohorts are described in Table 1. Our strategy is outlined in Fig 1. We began with a large cohort of 239 human subjects free of neurological disease ages 25–97 yr, the CommonMind (CM) cohort, to determine transcripts that increase or decrease with age. Subject-level characteristics of the CM cohort can be found in Table S1. Note that all cohorts in this study use brain tissue from the prefrontal cortex (PFC) because PFC is highly affected by aging (Glorioso & Sibille, 2011) and a variety of neurodegenerative diseases but shows little to no significant neuronal death with age (Haug et al, 1984; Morrison & Hof, 1997; Yankner et al, 2008; Glorioso & Sibille, 2011), unlike many other regions of the brain. Therefore, expression studies in PFC should be minimally confounded by changing cell-type numbers. We found that ~7% of all transcripts differ by age in a monotonic way across the entire cohort. Thus, we identified 834 transcripts that decrease and 537 transcripts that increase (Table S2). Importantly, our transcriptionally defined brain aging signature begins in early adulthood and progresses linearly thereafter. We then used a computational model (elastic net regression controlling for potential sources of noise such as sex and RNA quality) to calculate a molecular age for each brain.

The CM cohort showed a high correlation of molecular age and chronological age at the time of death (Fig 2A). We next used three other cohorts, PsychEncode (PE) (N = 216), GTEx (N = 87) (The GTEx Consortium, 2015), and BrainCloud (BC) (N = 127), to validate the predictive power of the method developed in the CM cohort. We were able to correlate molecular age and chronological age well in all cohorts (Fig 2A–H). It is notable that the older subjects in the CM and PE cohorts appeared to have a reduction in the Pearson correlation coefficient (R value) relating molecular and chronological age compared with younger subjects (Fig 2C and E). A large cohort (N = 430) of subjects (67–108 yr) that comprises subjects from the Religious Order Study and Rush Memory and Aging Project (ROS-MAP) was also tested (Bennett et al, 2012a, 2012b) (Table S3). This cohort is much older than the others, which may explain the somewhat lower R value for chronological versus

**Table 1. Cohort characteristic.**

| Cohort | No. subjects | No. male | No. Caucasians | No. African American | No. Hispanic | No. Asian | Mean age | Mean PMI | Mean RIN | Mean pH | Mean Education |
|---|---|---|---|---|---|---|---|---|---|---|---|
| CM | 239 | 147 | 179 | 39 | 17 | 3 | 63 yr | 15 h | 7.7 | 6.7 | N/A |
| BrainCloud | 127 | 85 | 58 | 69 | 4 | 3 | 48 yr | 35 h | 8.1 | N/A | 10 yr |
| PE | 216 | 145 | 211 | 0 | 2 | 2 | 70 yr | 11 h | N/A | 6.5 | N/A |
| GTEx | 87 | 62 | 76 | 11 | 0 | 0 | 58 yr | 14 h | 7.4 | N/A | N/A |
| ROS-MAP | 438 | 163 | 438 | 0 | 0 | 0 | 89 yr | 7 h | 7.2 | N/A | 17 yr |

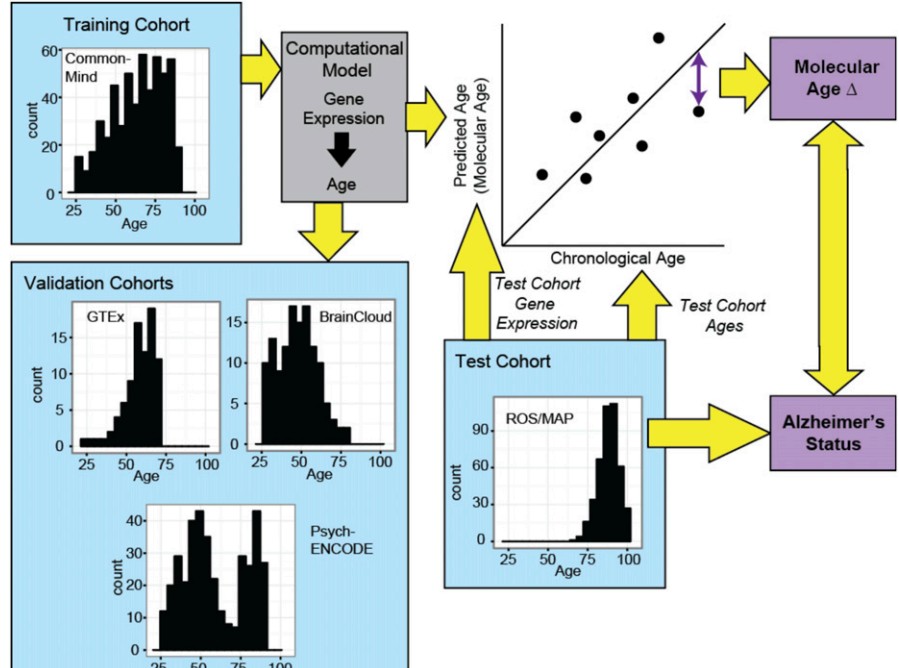

**Figure 1. Schematic of our approach.**
Age-sensitive transcripts are determined from a large training cohort of disease-free brains. These are used to create "molecular ages" and applied to four additional cohorts, one of which includes subjects with neurodegenerative diseases. The deviation of molecular age from chronological age (Δ age) is used to test associations of diseases and phenotypes with rates of aging of the brain.

molecular age (Fig 2H). Note this correlation is still highly significant ($P = 10 \times 10^{-8}$). Older subjects may generally show a reduction in correlation due to a time-dependent divergence in differences in aging rates among individuals in the cohorts. It could also be due to a "survivor effect," that is, the oldest subjects are more successful agers than their younger counterparts and have relatively younger brains.

A proxy for the rates of brain aging was obtained as the difference between molecular age and chronological age for each brain, termed Δ age (see Fig 1). We, thus, calculated Δ age for each brain as the difference between each individual data point and the regression line in all cohorts. Consistent with a survivor effect, the oldest subjects in CM (age > 60 yr) show a significant inverse relationship between Δ age and chronological age (Fig S1A), which is not present in subjects younger than 60 yr (Fig S1B). This is consistent with the reported prediction of all-cause mortality by methylation-based transcriptional age in blood (Marioni et al, 2015). Interestingly, as might be predicted based on differences in lifespan, men had significantly older Δ ages than women in the CM cohort (Fig S2).

## Comparison of transcription-based and methylation-based assignments of ages

Methylation data were available for subjects in the BC and ROS-MAP cohorts, and we used these as a second way to assign molecular age (Horvath, 2013). As shown in Fig 3A and B, there was a strong correlation between molecular and chronological age in BC (R = 0.98), and ROS-MAP (R = 0.67, $P < 2.2 \times 10^{-16}$), as had been previously shown (Levine et al, 2015). We next wondered whether the transcriptional and methylation methods would reveal similar deviations from

normal aging by comparing Δ age in these two cohorts as determined by both methods. Indeed, there was also a highly significant correlation (R = 0.69, $P = 1.2 \times 10^{-11}$) in BC and in ROS-MAP (R = 0.43, $P < 2.2 \times 10^{-16}$) between the two methods (Fig 3C and D). It is unlikely that the two methods reveal the same molecular events because methylation data apply to many different tissues showing highly variable transcriptional changes with aging (and see below). Indeed, we confirmed that the DNA methylation markers were not proximal to age-regulated transcripts (data not shown), as has been observed in other tissues (Horvath, 2013).

## Analysis of aging-sensitive transcripts

Transcripts which have levels that correlate with age in the CM cohort and PE cohort are illustrated in a heat map in Fig 4A and listed in Table S2. Each column represents one subject and shows the 537 transcripts that increase with age (top in red) or the 834 transcripts that decrease with age (bottom in blue). It may be observed that most transcripts show continuous incremental differences with the age of the subjects, suggesting that brain aging is a continual process from early in adulthood. This finding along with the published data on PFC (Haug et al, 1984; Morrison & Hof, 1997; Yankner et al, 2008; Glorioso & Sibille, 2011) argues that the transcriptional changes are not simply due to neuronal loss and a correspondingly higher glial composition. It is also evident that some brains appear exceptional, either slowed or advanced for aging compared with the average. These exceptional cases show the highest absolute values of Δ age in our analysis described above.

We grouped the aging-sensitive transcripts functionally by Ingenuity software. Interestingly, for transcripts that are lower in

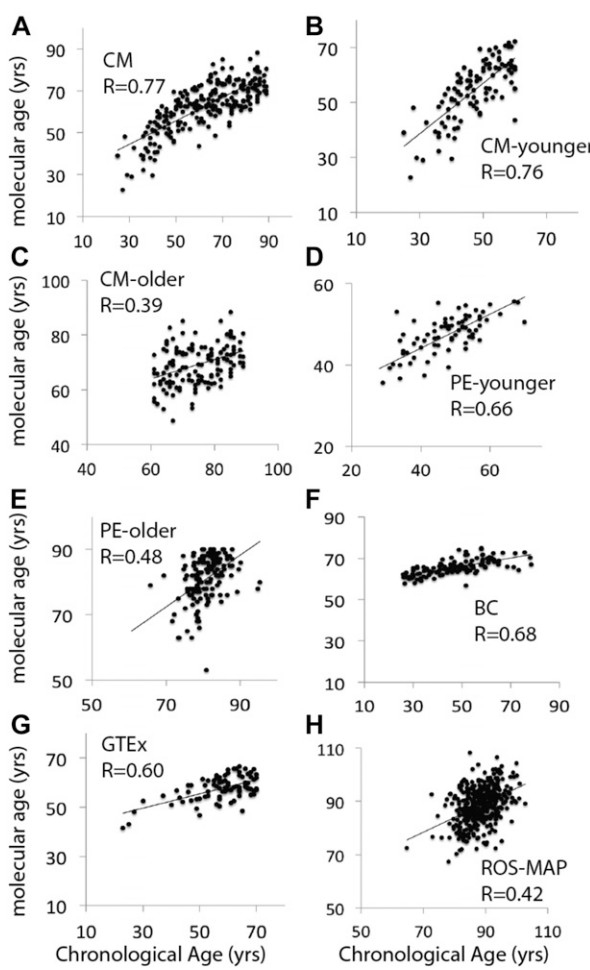

**Figure 2. Molecular ages were significantly predictive of chronological ages in all cohorts.**
**(A–H)** P-values corresponding to the R-values depicted are (A) $P = 8.3 \times 10^{-49}$. (B) $P = 2.1 \times 10^{-6}$. (C) $P = 3.2 \times 10^{-20}$. (D) $P = 1 \times 10^{-10}$. (E) $P = 1.2 \times 10^{-9}$. (F) $P = 1.3 \times 10^{-18}$. (G) $P = 9.8 \times 10^{-10}$. (H) $P = 1.7 \times 10^{-19}$. R-values were determined by Pearson correlation. The age-sensitive transcripts used to predict molecular ages are listed in Table S2. Note that the PE cohorts were analyzed as two separate cohorts because of confounding variables from different collections (see the Materials and Methods section).

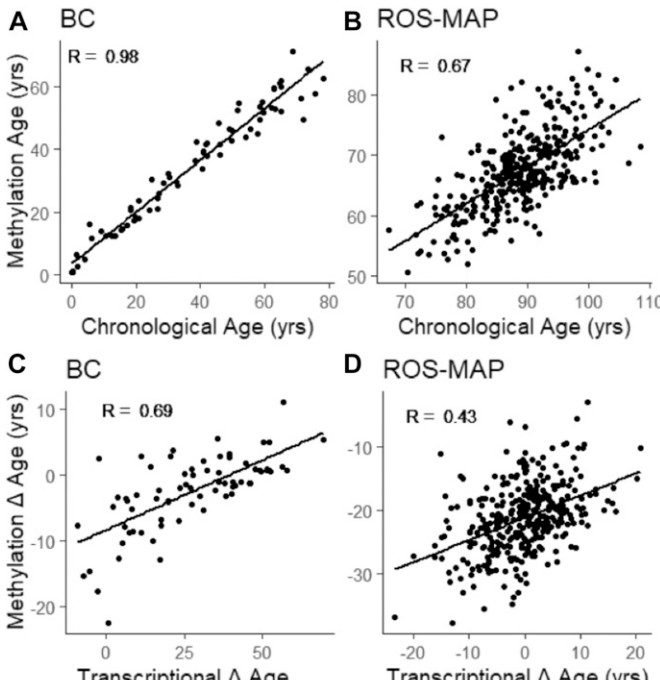

**Figure 3. Methylation ages and Δages in BC and ROS-MAP**
**(A, B)** Methylation ages are predictive of chronological ages in BrainCloud (A) and ROS-MAP (B). **(C, D)** Methylation and molecular delta ages correlate in BrainCloud (C) and ROS-MAP (D). R values were determined by Pearson correlation.

older people, neurological disease genes were a top disease category (Fig 4A). This included genes that relate to PD ($P = 2 \times 10^{-3}$), Tauopathy ($7 \times 10^{-3}$), Huntington's ($P = 2 \times 10^{-11}$), ALS ($4 \times 10^{-3}$), basal ganglia disorders ($1 \times 10^{-10}$), and other neurodegenerative disorders as well as to psychiatric disorders, including anxiety ($P = 1 \times 10^{-4}$), depression ($2 \times 10^{-4}$), bipolar disorder ($1 \times 10^{-3}$), and schizophrenia ($2 \times 10^{-7}$). More specific pathway categories for down-regulated genes include "glutamate signaling," "dopamine feedback in cAMP signaling" (not shown, $P = 1 \times 10^{-5}$), and "rho GTPases" (Fig 4A). The rho-GTPases are particularly interesting because they are known regulators of synaptic spine formation and actin cytoskeletal dynamics (Morrison & Baxter, 2012; Lefort, 2015). Thus, these changes are consistent with earlier findings showing deficits in synaptic function and neuronal signaling.

Interestingly, the other three categories of down-regulated genes are "mitochondrial dysfunction," "oxidative phosphorylation" (not shown, $P = 2 \times 10^{-5}$), and "sirtuin signaling." One exciting possibility is that defective sirtuin function contributes to mitochondrial and oxidative phosphorylation defects, which then impair neuronal function. However, we cannot discern whether mitochondrial dysfunction causes defects in neuronal function, neuronal functional defects cause mitochondrial defects, or the two are causally unlinked.

For up-regulated genes, cell morphology, immune cell trafficking, cancer, and cell-to-cell signaling were top categories. Also among up-regulated categories are pathways involved in inflammation and DNA damage/cancer, consistent with earlier findings (Yankner et al, 2008; Glorioso & Sibille, 2011) and illuminating the deteriorating environment of the aging brain.

Next, we mapped our transcript data onto the various cell types of the brain, in part to gain additional insight to whether transcriptional changes might be partially confounded by small changes in cellular composition. A recent study has identified consensus brain cell type–specific transcriptional markers based on the overlap of five murine and human single cell RNA-seq studies (McKenzie et al, 2018). We first grouped these transcripts as "neuronal" or "glial" (encompassing astrocytes, oligodendrocytes, and microglia) and queried the fraction that decreased (Fig 4B) or increased (Fig 4C) with aging. If there were significant loss of neurons in the PFC with age or gain of glia in the CM dataset, it might be expected that all neuronal-specific transcripts would decrease with age and that all glial-specific transcripts would increase with age (with some small margin of error for statistical chance). As

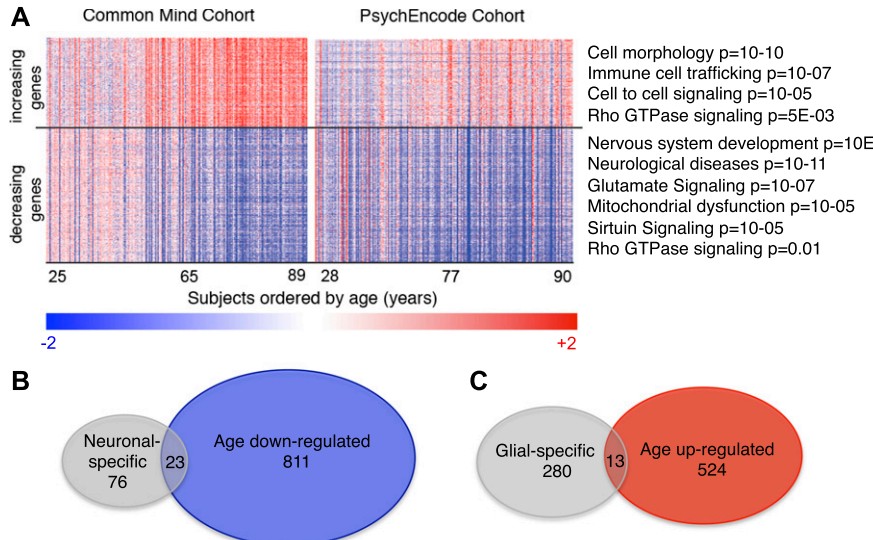

**Figure 4. Characterization of age-sensitive transcripts and molecular ages.**
**(A, B)** The 537 increasing and 834 decreasing age-sensitive transcripts are visualized in the heat map (A). Top Ingenuity functional categories are shown for increasing or decreasing transcripts (A). **(B, C)** Venn diagrams show the intersection of age down-regulated transcripts and neuronal-specific transcripts (B) and age up-regulated transcripts and glial-specific transcripts (C).

evident in the Venn diagrams, this was not the case as only a small fraction of the neuronal- or glial-specific transcripts changed with age, providing further evidence for the surmise that changes in cellular composition occur minimally in the aging PFC and do not account for the aging-sensitive transcripts. Furthermore, the methylation clock, which is likely independent of transcription and thus not similarly susceptible to cellular composition changes, was totally consistent with the transcriptional gauge. We, thus, conclude that changes in cellular composition are unlikely to explain the aging-regulated transcripts.

## Transcriptional Δ age associates with AD, PD, and other phenotypes

Importantly, ROS-MAP subjects had been followed longitudinally by medical examination for a variety of diseases and phenotypes, including AD, dementia, motor, and cognitive function, and their brains were characterized for disease-related pathology after death. *APOE* genotype data were also available for ROS-MAP subjects. We first investigated the relationship of Δ age in ROS-MAP as applied to 38 clinical and pathological phenotypes. The average age of this cohort at the time of death was 89 yr and about 2/3 of subjects had at least one indication of some form of dementia. We covaried for *APOE ε4* when assessing the relationship of Δ age to variables and likewise covaried for Δ age when assessing the relationship of *APOE ε4* to variables to isolate their effects. *P*-values were corrected for multiple testing. We found that a positive Δ age (older molecular age compared with chronological age) significantly associates with risk of clinical diagnosis of AD, with AD subjects showing significantly older molecular aging than non-disease controls (Fig 5A). Corroborating this finding, subjects with older Δ age values also performed worse on the clinical Alzheimer's dementia mini mental examination (Fig 5B). Moreover, the ROS-MAP brains were also quantified for levels of a pathological marker of AD, tangles, which was significantly and positively associated with a positive value of Δ age (Fig 5C). However, *APOE ε4* had much more

significant relationship than Δ age to the pathological measures of AD, amyloid, and tangles (Table 2), which may represent a mechanistic difference between the risk factors.

A positive Δ age was also associated with Parkinsonian score assessed across all 430 ROS-MAP subjects, which is a composite score of clinical signs of PD comprising rigidity (muscle stiffness), tremor (involuntary oscillation of limbs), gait (shortened, shuffling walking), and bradykinesia (Fig 5D) and with rigidity and gait independently (Fig 5E and Table 2). Likewise, positive Δ age was also significantly associated with the presence of PD pathology (*P* = 0.006, Fig 5F). We did not see an association with PD diagnosis, but this may be because we were less powered by number of PD-diagnosed subjects in the cohort (n = 31).

The strongest association of positive Δ age was with global cognition slope (*P* = 5 × 10⁻⁵, Fig 5G), which is the composite rate of cognitive decline over time for five different cognitive domains (episodic memory, visual-spatial ability, perceptual speed, semantic memory, and working memory). These data include all 430 ROS-MAP subjects, although strong associations were found individually in AD and disease-free subjects (Table S4). Each of the above cognitive domains was also individually significant for association with positive Δ age (Fig 5H and Table 2).

Because our goal was to probe any association between aging and risk alleles in AD, we tested for an association between *APOE ε4* and Δ age by stratifying the entire ROS-MAP population into groups with 0, 1, or 2 *APOE ε4* alleles. There was a weak association found between Δ age and one copy of *APOE ε4* (Fig 5I). As expected, the *APOE ε4* allele showed a highly positive association with AD- and cognition-related measures (Table 2).

All told, our findings suggest that positive Δ age has a large impact on risk for a variety of common late-life diseases and impairments, and to a similar extent as *APOE ε4*. Whereas most brain-related diagnoses and phenotypes associated are significantly with Δ age, peripheral phenotypes such as cardiovascular phenotypes and pathology, thyroid disease, and cancer did not (Table 2). This dichotomy may reflect the fact that our methods used

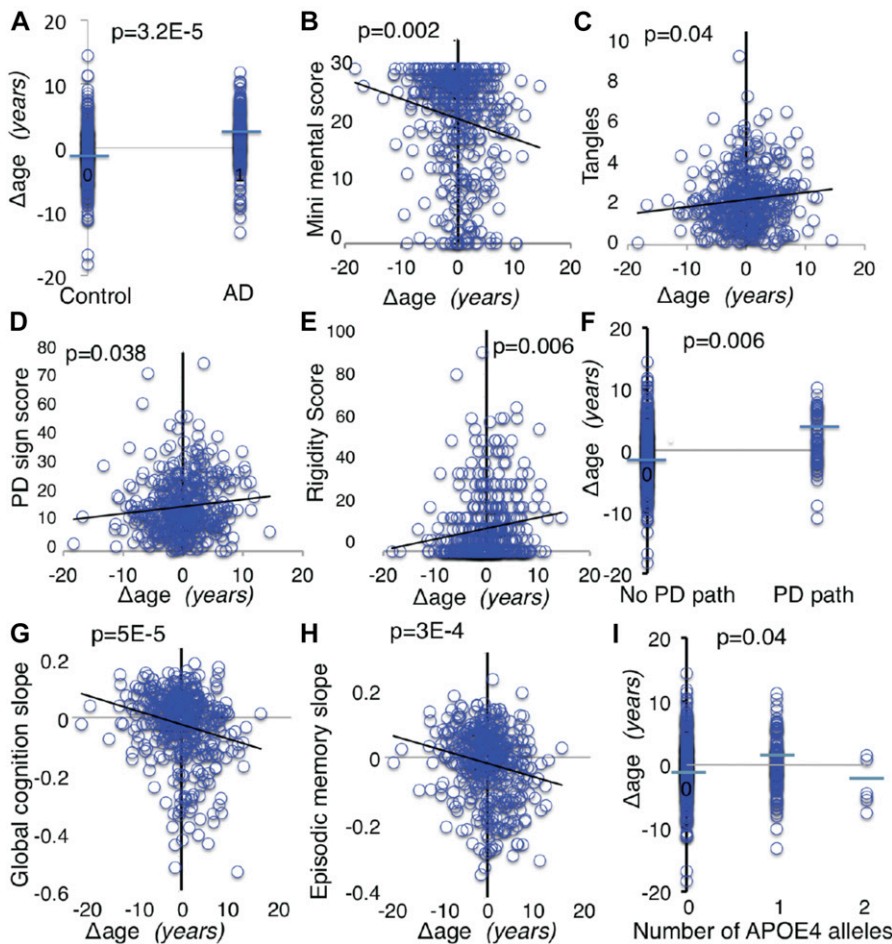

**Figure 5. Relationship of Δ age to clinical variables. (A–I)** Representative plots are shown using raw data (A–I). *P* values were determined by linear regression with relevant covariates. (A) Postmortem final clinical diagnosis of AD, (B) mini mental examination score, (C) tangles density, (D) Global PD score, a composite score for 4 signs: tremor, rigidity, bradykinesia, and gait, (E) rigidity score, (F) PD pathology is present if Lewy bodies are present and there was moderate-to-severe neuronal loss in the substantia nigra, (G) Global cognition slope, a composite slope of the longitudinal changes over time in five domains of cognition: working memory, visual-spatial ability, perceptual speed, episodic memory, and semantic memory, (H) change in episodic memory over time, and (I) association of Δ age & *APOE ε4*.

to calculate Δ age are specific to brain, or might indicate that different tissues age at different rates in the same person (see the Discussion section).

### Comparing transcriptional and methylation Δ ages in relation to AD and related variables

We used the subset of ROS-MAP subjects that had both methylation and transcriptional data available (n = 336) to directly compare the impact of transcriptional Δ ages to methylation Δ ages (Table S5). As with the 430 subjects shown above, we found that older transcriptional Δ ages continued to be significantly associated with Alzheimer's diagnosis ($1.3 \times 10^{-5}$), faster global cognitive decline ($P = 1.8 \times 10^{-5}$), episodic memory decline ($P = 1.2 \times 10^{-4}$), visual-spatial ability decline ($P = 1.3 \times 10^{-4}$), perceptual speed decline ($P = 5 \times 10^{-5}$), working memory decline ($P = 0.02$), greater PD signs ($P = 0.01$), more severe dementia grade ($P = 7.6 \times 10^{-0.05}$), tangles ($P = 1.0 \times 10^{-5}$), and Lewy body pathology ($P = 0.005$) (Table S5). With the exception of dementia grade ($P = 0.006$), these variables were not significantly associated with methylation Δ ages (Table S5). However, depression score ($P = 0.03$), amyloid ($P = 0.05$), Parkinsonian gait ($P = 0.03$), bradykinesia ($0.05$), and PD pathology ($0.001$) were associated with methylation Δ ages. These findings suggest that both methods are useful predictors, and that the transcriptional gauge may have a broader reach in associating with the risk of brain dysfunctional endpoints in aging.

### Transcriptional Δ age and *APOE ε4* are synergistic risk factors for AD

To further explore the relationship of Δ age and *APOE ε4* to each other and AD, we calculated the odds of having a clinical diagnosis of AD with respect to each. We binned Δ age into younger (−1 SD from the mean or ~ −5 molecular years), neutral (−5 y to +5 y), or older (+1 SD from the mean or +5 molecular years), and combined subjects that were homozygous for *APOE ε4* with those that were heterozygous because there were only four homozygous subjects. We found that Δ age and *APOE ε4* are synergistic risk factors for AD (Fig 6A). For example, subjects who are +5 y and bear one or two *APOE ε4* alleles have more than 5× the average odds of having AD. However, subjects who are −5 molecular years and with one or two *APOE ε4* alleles have no elevated chance of AD compared with subjects of average molecular age with no *APOE ε4* alleles. These findings suggest that younger Δ age can protect against *APOE ε4* alleles. In summary, our findings suggest that *APOE ε4* and older Δ age contribute synergistically to risk of AD and a variety of

**Table 2.  False discovery corrected *P*-values obtained from regression of either Δ age (main method) or *APOE ε4* with indicated disease or aging variables of interest.**

| Variable | Δ Age | *APOE ε4* |
|---|---|---|
| | *P*-value | *P*-value |
| ΔGlobal cognition/yr | $5.1 \times 10^{-5a}$ | $1.9 \times 10^{-5a}$ |
| ΔEpisodic memory/yr | $2.9 \times 10^{-4a}$ | $2.5 \times 10^{-6a}$ |
| ΔVisual-spatial ability/yr | $0.003^a$ | $0.02^a$ |
| ΔPerceptual speed/yr | $8.7 \times 10^{-5a}$ | $0.02^a$ |
| ΔSemantic memory/yr | $5.1 \times 10^{-5a}$ | $1.6 \times 10^{-5a}$ |
| ΔWorking memory/yr | $0.03^a$ | $0.006^a$ |
| Global cognition level | $3.2 \times 10^{-5a}$ | $2.4 \times 10^{-6a}$ |
| Episodic memory level | $9 \times 10^{-5a}$ | $2.2 \times 10^{-6a}$ |
| Dementia grade | $0.006^a$ | $7.6 \times 10^{-5a}$ |
| AD clinical diagnosis | $3.2 \times 10^{-5a}$ | $7.9 \times 10^{-4a}$ |
| Mini mental examination score | $0.0017^a$ | $1.7 \times 10^{-4a}$ |
| Depression score | 0.69 | 0.85 |
| General pathology | $0.035^a$ | $5.2 \times 10^{-10a}$ |
| Plaque level | 0.06 | $8.1 \times 10^{-8a}$ |
| Tangles level | $0.04^a$ | $5.4 \times 10^{-7a}$ |
| Amyloid level | 0.15 | $5.4 \times 10^{-7a}$ |
| Amyloid angiopathy | $0.07^a$ | $9.4 \times 10^{-7a}$ |
| PD diagnosis | 0.88 | 0.28 |
| PD sign score | $0.038^a$ | $0.02^a$ |
| Gait | $0.031^a$ | $0.02^a$ |
| Bradykinesia | 0.41 | $0.02^a$ |
| Rigidity | $0.0059^a$ | $0.035^a$ |
| Tremor | 0.86 | 0.17 |
| PD Pathology | $0.006^a$ | 0.88 |
| Lewy body pathology | $0.02^a$ | 0.69 |
| Stroke diagnosis | 0.56 | 0.40 |
| "Heart problem" history | 0.17 | 0.90 |
| Hypertension at baseline | $0.003^b$ | 0.67 |
| Arteriolar sclerosis | 0.75 | 0.84 |
| Cerebral infarction gross | 0.47 | 0.40 |
| Cerebral infarction micro | 0.33 | 0.58 |
| Cancer history | 0.91 | 0.95 |
| Thyroid disease history | 0.08 | 0.60 |
| Smoking (lifetime pack-years) | 0.75 | 0.46 |
| APO *ε4* alleles | $0.035^a$ | 0 |
| Δ Age | 0 | $0.035^a$ |

[a]Indicate significantly increased risk with older Δ age or greater *APOE ε4* alleles.
[b]Indicates the inverse relationship.

age-related neurological diseases and dysfunction (Fig 6B), and *APOE ε4* is unlikely to increase risk by simply increasing the rate of brain aging.

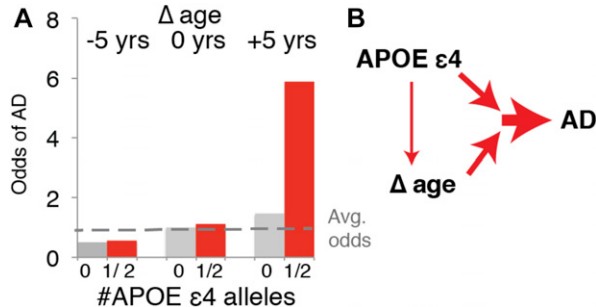

**Figure 6.  *APOE ε4* and Δage are synergistic risk factors for AD**
**(A)** Odds ratio of Δ age and *APOE ε4* for having AD diagnosis. **(B)** Model showing synergistic effects of Δ age and *APOE ε4* on AD.

## Discussion

In this study, we have carried out a detailed analysis of brain aging to ascertain how it interacts with other risk factors for neurodegenerative diseases. We used transcriptomic analysis to assign a molecular age to PFC samples from a series of cohorts from brain banks and calculated the deviation of molecular age from chronological age (Δ age) as a proxy for the rate of aging. These methods were developed by entraining the algorithm on one cohort of disease-free brains and applying them to four additional cohorts. Our method showed a highly significant correlation between molecular and chronological age for all cohorts. We then examined association of aging rates (Δ age) determined in nondisease brains with AD brains in the ROS-MAP cohort, which is highly enriched in AD, and made several important findings. First, a positive Δ age (rapid aging compared with average across all brains) associates with risk for AD with a high significance. Second, *APOE ε4*, the strongest genetic risk factor for sporadic AD, also strongly associates with AD risk in the same cohort, as expected. Third, a rapid aging rate and *APOE ε4* are synergistic risk factors. Brains with the slowest aging are strongly protected against having the *APOE ε4* allele and brains with the fastest aging have a greatly elevated AD risk when combined with *APOE ε4*. Our findings thus suggest that AD can be induced by the simultaneous occurrence of two risk factors and that interventions against either one might protect against the disease. It is intriguing that imaging studies report an effect of *APOE ε4* on brain structure as early as infancy (Dean et al, 2014), suggesting that *APOE ε4* alleles are pathological and not simply drivers of premature aging. Our data show a weak association between *APOE ε4* and brain aging. The DNA methylation "clock" (Horvath, 2013) correlated well with the transcriptome gauge but was not as broad in predicting risk of brain dysfunctional endpoints in aging.

Our analysis also provides several molecular insights into brain aging in the neurologic disease-free cohorts. First, aging is associated with changes in transcripts affecting rho-GTPases, which are associated with synapse formation and actin cytoskeleton dynamics in axons (Lefort, 2015). Other top categories of transcripts reduced in the aging brain encode GTPase inhibitors and other synaptic function-related proteins. Deficits in all of these transcripts are exacerbated in the much older ROS-MAP cohort. Thus, our findings are consistent with earlier findings of synaptic deficits

in the aging brain (Lefort, 2015) and suggest that the transcriptional deficits we observe may trigger the defect in synapses. This is also consistent with a study showing decreases in synaptic genes in AD subjects versus control subjects that occur at the onset of neuropathology and are exacerbated in subjects that are *APOE ε4* positive (Bossers et al, 2010). Second, there is a decrease in transcripts associated with neuronal signaling and mitochondrial/ sirtuin function. In particular, we find a reduction in glutamate receptor signaling, which may partly explain the strong association observed between the rate of normal brain aging and cognitive decline. The mitochondrial/sirtuin category included many genes involved in electron transport and ATP synthesis. This novel finding suggests an interplay between the sirtuin/mitochondrial pathway, neuronal signaling, and synaptic function in aging. It remains to be seen if one of these categories is the causal event in driving the transcriptional down-regulation and presumed deterioration in neurons, but it is possible that intervention strategies, such as sirtuin activation, may be protective against neurodegenerative diseases. Third, there is an up-regulation in inflammatory pathways, which is indicative of glial activation and consistent with earlier studies. Fourth, there is a coordination in the age differences of all of the aging-sensitive transcripts, suggesting that segmental aging does not occur, at least in the PFC. Fifth, although the rate of brain aging strongly correlates with AD and other brain maladies, there is no association with cardiac disease or cancer. This interesting finding raises the possibility that the aging of different tissues is not coordinated with brain aging in an individual. Indeed, our data do not directly address whether rates of aging are coordinated across different regions of the brain (although see discussion of PD below). Sixth, there is no association between brain aging and a history of smoking, which might have been expected (Mayeux & Stern, 2012).

We do not believe that the changes we observe in PFC aging are simply due to neuronal loss or glial gain. Indeed, the transcriptional changes we used to define brain aging begin early in adulthood, before any significant neuronal loss would occur. In addition, there is little change in the CM or PE cohorts in transcripts defined as neuronal-specific or glial-specific (McKenzie et al, 2018), suggesting there is not a significant loss of neurons in PFC, consistent with earlier data (Haug et al, 1984; Morrison & Hof, 1997; Yankner et al, 2008; Glorioso & Sibille, 2011). Finally, assigning molecular by the transcriptome gauge correlated well with assignment by the DNA methylation clock, the latter of which is predictive of age for many tissue types (Horvath, 2013). We thus conclude that aging-promoted expression changes within the brain are not explained by neuronal loss or glial gain.

In the ROS-MAP cohort, we also observed a significant association between the rate of brain aging and other neurological disorders. This observed association with Parkinson's symptoms is perhaps more surprising than the AD association because the dopaminergic neurons affected in PD are in a distinct brain region and have a distinct function compared with cortical neurons used to entrain the aging algorithm.

We also found in both control and disease ROS-MAP subjects, a very strong association between rapid brain aging and cognitive decline. This association indicates that brain aging predisposes to loss of cognitive functions and serves as a strong validation that our assignment of molecular ages is robust.

The question arises whether fast brain aging causes AD or AD somehow triggers the rapid aging. The fact that Ingenuity Analysis associated many of the aging-sensitive transcripts with neurodegenerative diseases underscores the complexity of determining cause and effect. Several factors lead us to favor a model that rapid brain aging is a cause of AD and not a result. First, brain aging was defined in the CM and PE cohorts, comprising subjects with no disease diagnoses and in which brains were judged to be free of pathology. Second, aging-sensitive transcripts used to gauge molecular brain age changed from early ages and in a continuous way, well before any disease processes could have set in. Third, the association of older molecular brain age with a variety of diseases suggests a causal link between brain aging and AD rather than accelerated aging being a consequence of the pathological brain. For example, if the aging were an effect, then we would have to conclude that both AD pathology centered in the cortex and PD pathology centered in the dopaminergic neurons could both somehow speed up the aging of the PFC. Furthermore, although *APOE ε4* strongly associated with AD pathology, there was a much weaker link between AD pathology and Δ age, suggesting that *APOE ε4* and Δ age may at least in part have separate mechanisms. Fourth, Δ age strongly associated with the rate of cognitive decline in subjects without diseases.

Our findings suggest that slowing brain aging might delay or favorably slow progression of AD, PD, cognitive decline, and potentially other neurological conditions of old age. Slowing molecular aging may be most effective as a preventative strategy before irreversible neuronal loss has occurred. In this regard, the detailed analysis of molecular brain aging may lead to specific genes and pathways that regulate the rate of aging and offer therapeutic targets for intervention to impact a broad spectrum of neurological diseases and deficits.

# Materials and Methods

### Cohorts

We used five different cohorts of human PFC samples. These five cohorts have been described in previous publications, CM (Fromer et al, 2016), PE (Akbarian et al, 2015), BrainCloud (Colantuoni et al, 2011), GTEx (The GTEx Consortium, 2015), and ROS-MAP (Bennett et al, 2012) and below. Summary statistics for cohort characteristics can be found in Table 1.

### *CM*

The CM cohort (Fromer et al, 2016) contains control subjects across the dorsolateral PFC (DLPFC; Brodmann areas 9 and 46) from brain banks at the Icahn School of Medicine at Mount Sinai, the University of Pennsylvania, and the University of Pittsburgh. These subjects are free of neurological disease from medical history, direct clinical assessments, interviews of family members or care providers, and pathological report. Brains were examined grossly for infarcts and gross pathology. Frontal lobes, hippocampus, entorhinal cortex, and cerebellum were stained with H&E. Frontal lobes, hippocampus, and entorhinal cortex were additionally stained with Bielschowsky

silver stain, and frontal lobes and hippocampus/entorhinal cortex were stained with β-A4 and α-synuclein immunostain, respectively. Cases were excluded if they had history or report of psychiatric or neurological disease or neuropathology related to AD and/or PD, acute neurological insults (anoxia, strokes, and/or traumatic brain injury) immediately before death, or were on ventilators near the time of death.

More details can be found in Fromer et al (2016), but in brief, total RNA was isolated from ~50 mg homogenized tissue in Trizol using the RNeasy kit and was processed in batches of 12. Samples with RNA integrity number (RIN) < 5.5 were excluded from the study. Samples were prepared for RNA sequencing using the Ribo-Zero Magnetic Gold Kit (Cat # MRZG12324; Illumina/Epicenter) to enrich for polyadenylated coding RNA and noncoding RNA and the TruSeq RNA Sample Preparation Kit v2 (RS-122-2001-48 reactions) in batches of 24 samples. A pool of 10 barcoded libraries were layered on a random selection of two of the eight lanes of the Illumina flow cell bridge amplified to ~250 million raw clusters. 100-bp paired-end reads were obtained on a HiSeq 2500.

### PE

We used control subjects of both sexes from the BrainGVEX cohort of the PE database. These were analyzed separately because of heterogeneity in RNA quality and age variables between the cohorts that could potentially confound analyses. These subjects were free of neurological diseases by pathological report and medical history. BrainGVEX uses postmortem human brain material from two institutes across multiple brain collections. Fresh-frozen brain samples are from four collections of the Stanley Medical Research Institute (SMRI) and Banner Sun Health Research Institute (BSHRI). The SMRI collection contains human postmortem brain samples from four brain collections: the Neuropathology Consortium, Array Collection, New Collection, and Depression Collection. These specimens were collected, with informed consent from next-of-kin, by participating medical examiners. Diagnoses of unaffected controls were based on structured interviews by a senior psychiatrist with family member(s) to rule out Axis I diagnoses. Exclusion criteria included the following: 1. significant structural brain pathology on postmortem examination by a qualified neuropathologist or by premortem imaging; 2. history of significant focal neurological signs premortem; 3. history of a CNS disease that could be expected to alter gene expression in a persistent way; 4. documented IQ < 70; 5. poor RNA quality; and 6. substance abuse within 1 yr of death or significant alcohol-related changes in the liver.

PE BrainGVEX also contains human postmortem brain samples from the BSHRI Brain Donation Program (Beach et al, 2008). Eligibility criteria for the program includes that subjects must consent to annual clinical assessments at SHRI. In addition, at least 2 yr of the applicant's private medical records must be received and reviewed by Brain Donation Program staff before acceptance. All enrolled subjects or legal representatives sign an Institutional Review Board–approved informed consent form allowing both clinical assessments during life and several options for brain and/or bodily organ donation after death. Between 1987 and 1995, brain donors did not receive formal neuropsychological testing. Their mental status was determined by requisitioning medical records

from their primary care physicians, neurologists, psychologists, and psychiatrists, and through telephone interviews with family members and caregivers, both at the time of enrollment and in the immediate postmortem period. In 1996, a clinical psychologist was hired and from then onwards, a standardized neuropsychological screening assessment has been administered to most of the subjects enrolled in the Brain Donation Program. Gross neuropathologic examinations on brain external surfaces, coronal cerebral slices, and parasagittal cerebellar slices were performed by the neuropathologist.

Total RNA was isolated by organic extraction (SMRI) or miRNeasy Mini Kit (BSHRI). To pass QC to library generation, RNA must have concentration of ≥100 ng/uL assayed by Qubit 2.0 RNA BR Assay or Xpose, and RIN score ≥ 5.5 assayed by Agilent Bioanalyzer RNA 6000 Nano assay kit. All total RNA from both SMRI and BSHRI collections were processed into rRNA-depleted stranded libraries for sequencing on the Illumina HiSeq2000 using the TruSeq Stranded Total RNA Sample Prep Kit with Ribo Zero Gold HMR (#RS-122-2301; Illumina). Libraries are sequenced on Illumina's HiSeq2000 on a high output flow cell for 100-bp PE sequencing. Libraries are three-plexed per lane to reach 40 M paired-end reads per library.

Fastq files go through adapter removal using cutadapt, and then the resulting adapter-trimmed FASTQ files are checked for quality using FastQC. A subset of 10,000 reads is used to estimate insert mean size and SD for use with Tophat. Tophat is used to align trimmed reads to the GENCODE19 reference (modified to include artificial ERCC RNA ExFold spike-in sequences). Expression level are then calculated using HTSeq and Cufflinks with custom scripts used to summarize the proportion of reads assigned to each RNA type (i.e., protein_coding, snoRNA, and rRNA). Parameters that account for unstranded read orientation for polyA and stranded read orientation for Ribo-Zero libraries were used for Tophat and Cufflinks. FASTQ files are trimmed for adapter sequence and base quality using cutadapt, then subject to FastQC quality checks. Alignment is carried out with STAR in two-pass mode to the GENCODE19 reference genome. The resulting alignments are then sorted and merged (if there were multiple pairs for a given sample) using NovoSort.

### BrainCloud

The BrainCloud cohort comprises brains that are free of neurological disease from medical history and pathological report. Neuropathological examination was performed in all cases by a board-certified neuropathologist. Brain sections through several cortical regions and the cerebellar vermis were examined microscopically, including the use of Bielschowsky's silver stain. Cases with cerebrovascular disease (infarcts or hemorrhages), subdural hematoma, neuritic pathology, or other significant pathological features were excluded from further study. Cases with acute subarachnoid hemorrhages that were directly related to the immediate cause of death were not excluded.

More details can be found here (Lipska et al, 2006; Colantuoni et al, 2011), but in brief, RNA was extracted from fresh frozen PFC and run on Illumina Human 49K Oligo array (HEEBO-7 set). To compare with the other datasets, we summarized the expression values at the gene level. BioMart (ENSEMBL version 73) was used to convert the given National Center for Biotechnology Information gene IDs to the ENSEMBL gene id. For every ENSEMBL gene id, we collapsed into

the gene level by taking the mean across all oligos that mapped to the gene. Data are available at http://BrainCloud.jmhi.edu/.

### GTEx
The GTEx cohort comprises brains that are free of neurological disease from medical history and pathological report. More details can be found here (The GTEx Consortium, 2015), but in brief, RNA was extracted from fresh frozen PFC and run using RNAseq. Data were summarized into reads per kilobase of transcript, per Million mapped reads (RPKM) values for GENCODE gene model. Data are available at the database of genotypes and phenotypes (dbGaP) access id phs000424.v6.p1. The GENCODE RPKM values were converted to ENSEMBL using BioMart (ENSEMBL version 73).

### ROS-MAP
The ROS-MAP cohorts are community based cohort studies of aging in which all participants are organ donors (Bennett et al, 2012). We used brains with transcriptomic data comprising subjects with and without a variety of clinical diagnoses and phenotypes. Cohort characteristics can be found in Tables 1 and S1. RNA was extracted from fresh frozen PFC and run using RNAseq. RNA integrity was 5.0–9.9 and postmortem intervals were 0–41 h. The dataset had already been assembled into RPKM values based on ENSEMBL gene ID. Data are available at https://www.synapse.org/#!Synapse:syn3219045.

## Experimental design and statistical analysis

### Data normalization
Further normalization and quality control procedures were applied to each of the datasets. First, outlier values for each gene were removed from each dataset (SD > 4). In each cohort, there were several samples for which a large number of genes were outliers. After confirming that those samples did not correspond to the oldest and youngest subjects, they were removed. Overall, these samples had lower than average RIN scores. After removing those samples, the remaining outlier values were imputed with a K-nearest neighbor algorithm (Hastie et al, 2016). To put the datasets on a comparable scale, we scaled the dataset by mean (0-normalized) and SD (normalized so the SD for each gene is 1).

### Calculation of molecular age
A combination of observed and unobserved factors can have large effects on broad patterns of gene expression and are often controlled for in gene expression studies (Stegle et al, 2012). In our study, we wanted to preserve any broad signature associated with age or molecular age, while still removing broad signatures associated with noise. To achieve this goal, we first cleaned the data noise variables, including RNA quality (RIN), sex, postmortem interval, reported race, and race calculated using principal components on genotypes (Purcell et al, 2007). They also included several variables specific to individual datasets: sample source (Brain-Cloud), cohort (ROS-MAP), study center (GTEx), and RNA preparation method (GTEx).

To estimate molecular age, we built a model linking gene expression to age using the cohort with the broadest distribution of ages and the most control subjects, CM. We used elastic net regression, which has emerged as the standard for predicting age based on DNA methylation data (Horvath, 2013). The model is regularized so that redundant features do contribute disproportionately, but flexible enough to be robust if several of the genes differ in their robustness across cohorts.

There are two parameters in the elastic net regression, $\alpha$ and $\lambda$. We chose parameters based on how well a model trained with the CM cohort could predict age in the PE cohort. The genes used for the model were those that were strongly significantly (Benjamini–Hochberg corrected $P < 0.001$, linear model) associated with age in the cohort. This lead to a total of 1,263 genes that were included in building the model. In each direction, the highest correlation between predicted and actual age occurred at $\alpha = 0.01$ (CM to PE R = 0.66). At $\alpha = 0.01$, the $\lambda$ that maximized this correlation between the two cohorts was 99. In total, 834 genes contributed to the model.

The estimated ages could not be directly used across cohorts because the dataset was mean-normalized and different cohorts have very different age ranges. To overcome this bias, we calculate a regression line between predicted and actual age and then estimated molecular age $\delta$ as the age difference between that regression line and predicted age.

### Calculation of methylation ages
The BrainCloud cohort includes 78 subjects who are aged 0–84 and have both transcriptome data as well as Illumina 27k methylation data from PFC samples. The ROS-MAP cohorts include 336 subjects ages 67–108 with both transcriptomic data as well as Illumina 450k methylation data from PFC samples. Horvath's methylation age is based on 353 probes common to both Illumina 27k and 450k methylation arrays from a range of tissues and cell types. Methylation ages were calculated using the method and R function described by Horvath (Horvath, 2013). Methylation Δ ages were defined as the calculated methylation age minus the chronological age.

### Calculation of cell type–specific transcript levels
Cell type–specific genes were obtained from a meta-analysis of three studies of sorted cell types in mice and two studies of sorted cell types in humans (McKenzie et al, 2018). Cell type–specific genes chosen were top 100 genes that showed highest enrichment for each cell type in all five studies, where enrichment is defined as expression of a gene in one cell type versus the expression of the same gene in all the other cell types. For the Venn diagram in Fig 4C, astrocytes, oligodendrocyte, and microglia were combined into the glial category.

### Calculation of correlation with clinical variables in ROS-MAP
For each continuous variable in ROS-MAP, Δ ages were run in a linear regression model subtracting APOE ε4, age, sex, race, and population principle components, study number, RNA integrity, and batch. Logistic regression was used in the case of binary variables such as AD diagnosis. To determine the relationship of APOE ε4 to each variable, the same procedure was used except instead of subtracting APOE ε4, we subtracted Δ age. P-values in all cases were corrected for multiple testing using the Benjamini–Hochberg method (Benjamini & Hochberg, 1995).

## Data access

All data used in this analysis have been previously described in publications and is available in online repositories (see the Materials and Methods section).

## Supplementary Information

## Acknowledgements

We thank Sharon Wu for help in analyzing data. This work was supported by a grant from The Glenn Foundation for Medical Research (LP Guarente) and AFAR (CA Glorioso). This research was conducted while CA Glorioso was an Ellison Medical Foundation/AFAR Postdoctoral Fellow.

### Author Contributions

CA Glorioso: conceptualization, formal analysis, supervision, funding acquisition, visualization, methodology, and writing—original draft, review, and editing.
AR Pfenning: conceptualization, formal analysis, supervision, investigation, visualization, and writing—review and editing.
SS Lee: formal analysis, visualization, and methodology.
DA Bennett: data curation, funding acquisition, validation, and writing—review and editing.
EL Sibille: data curation, funding acquisition, and writing—review and editing.
M Kellis: conceptualization and writing—review and editing.
LP Guarente: conceptualization, supervision, funding acquisition, and writing—original draft, review, and editing.

### Conflict of Interest Statement

LP Guarente is a founder of Elysium Health and Galelei BioSciences and on the SAB of GSK, Segterra, and Sibelius.

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
