## [Reviewer comments · Life Science Alliance]

Life Science Alliance

Rate of brain aging and APOE ϵ 4 are synergistic risk factors for Alzheimer's disease

Christin Glorioso, Andreas Pfenning, Sam Lee, David Bennett, Etienne Sibille, Manolis Kellis, and Leonard Guarente

DOI: <https://doi.org/10.26508/lsa.201900303>

Corresponding author(s): Leonard Guarente, MIT and Christin Glorioso, Massachusetts Institute of Technology

Review Timeline:

Submission Date:	2019-01-09
Editorial Decision:	2019-01-30
Revision Received:	2019-03-27
Editorial Decision:	2019-04-18
Revision Received:	2019-05-09
Accepted:	2019-05-10

Scientific Editor: Andrea Leibfried

Transaction Report:

January 30, 2019

Re: Life Science Alliance manuscript #LSA-2019-00303

Dr. Leonard Guarente
MIT
Dept. of Biology
MIT
77 Massachusetts Avenue
Cambridge, USA-Cambridge, MA 02139-4307 2215

Dear Dr. Guarente,

Thank you for submitting your manuscript entitled "Rate of brain aging and APOE ϵ 4 are synergistic risk factors for Alzheimer's disease" to Life Science Alliance. The manuscript was assessed by expert reviewers, whose comments are appended to this letter.

As you will see, the reviewers appreciate your analyses but think that more information on the individuals of each cohort needs to be provided. We would thus like to invite you to provide a revised version of your manuscript, addressing this point as well as the other issues raised by the reviewers. A non-linear model (see reviewer #2) should get considered and discussed as well, please.

Thank you for this interesting contribution to Life Science Alliance. We are looking forward to receiving your revised manuscript.

Sincerely,

- A letter addressing the reviewers' comments point by point.
- An editable version of the final text (.DOC or .DOCX) is needed for copyediting (no PDFs).
- High-resolution figure, supplementary figure and video files uploaded as individual files: See our detailed guidelines for preparing your production-ready images, <http://life-science-alliance.org/authorguide>
- Summary blurb (enter in submission system): A short text summarizing in a single sentence the study (max. 200 characters including spaces). This text is used in conjunction with the titles of papers, hence should be informative and complementary to the title and running title. It should describe the context and significance of the findings for a general readership; it should be written in the present tense and refer to the work in the third person. Author names should not be mentioned.

B. MANUSCRIPT ORGANIZATION AND FORMATTING:

Full guidelines are available on our Instructions for Authors page, <http://life-science-alliance.org/authorguide>

Reviewer #1 (Comments to the Authors (Required)):

Summary:

The manuscript "Rate of brain aging and APOEε4 are synergistic risk factors for Alzheimer's

disease" by Glorioso et al describes (1) the development of a measure of molecular brain age using transcriptome data from the postmortem brains of healthy human subjects aged 25-89 (N=239), and (2) the application of this measure to transcriptome data from the postmortem brains of 4 additional cohorts of healthy human subjects and a large group (N=438) of subjects with and without brain disease aged 67-108 at time of death. The authors found that their measure of molecular brain age was highly correlated with chronological age and DNA methylation age. The authors found that the group of subjects with older molecular brain ages (relative to chronological age) were enriched in subjects with Alzheimer's disease, Parkinson's disease, and cognitive decline. In contrast, the authors found that the group of subjects with younger molecular brain ages were depleted in subjects with Alzheimer's disease, even among those subjects with the APOEε4 allele.

Major Comments:

This study addresses an important question re: the relationship among age-related transcriptome changes, late-life brain disease/dysfunction, and APOEε4. The use of data from a large number of subjects from multiple brain tissue collections improves the generalizability of the findings. Integrating the analyses of molecular brain aging with that of APOE genotype adds value. The conclusions are well-supported but weaknesses exist that limit the strength of the conclusions that can be drawn.

In my view, the major weakness of the manuscript is the limited and non-uniform description of the cohorts. For each cohort, the approach to arriving at a diagnosis, or absence of a diagnosis, were different but this is not easy to discern from the methods section. The descriptions of the cohorts all contain different information, for example, RIN value (standard deviation) is reported for the Common Mind cohort but not for any of the others. The manuscript would benefit from a single table with the following values for each cohort: mean age, mean PMI, mean RIN, number of male and female subjects, number of subjects of each race. If this data is not available, then that should be explained.

Further, since the measure of molecular brain aging was developed using 239 subjects from the Common Mind cohort, the supplemental information should include subject-level information for age, PMI, RIN, sex, race, tissue bank.

Also for the supplemental information, delta age versus PMI and RIN for the 239 subjects from the Common Mind cohort should be plotted.

Additional Comments:

Was there any evidence suggesting that APOEε2 allele was associated with younger molecular brain ages?

Neuron cell death is repeatedly referred to as being a part of normal brain aging (p.7, p.11, p.21). It is my understanding that neuron cell death and normal brain aging are mutually exclusive phenomena. What is meant in these passages needs to be clarified.

What gene list was used as background for the Ingenuity analyses?

Pathway analyses were performed for transcripts that increase with age and transcripts that

decrease with age but not for all transcripts that change with age. What is the rationale for this?

Reviewer #2 (Comments to the Authors (Required)):

In this manuscript, Glorioso et al. developed a computational model to determine the molecular age (but not chronological age) of the individual brain transcriptome data, using transcriptomic datasets from a cognitively healthy cohort. After a validation using methylation dataset, the authors applied this model to investigate the relationship between molecular brain aging and clinical traits using ROS-MAP data. They found that advanced molecular aging of the brain is associated with major AD and PD-related phenotypes. Furthermore, they built a model in which interactions between APOE4, molecular aging and AD-related phenotypes were incorporated, and concluded that molecular aging and APOE4 are synergistic risk factors for AD. Linking molecular basis of aging with that of brain disorders is not conceptually novel; however, the use of multiple transcriptomic datasets and methylation data to define molecular aging signatures supports the rigor of the study. In addition, modeling between APOE4, molecular aging and AD-related phenotypes is of interest, as it might provide insight into the APOE4-driven AD-related pathogenesis. Overall, this is a well-written and interesting manuscript. To further strengthen this manuscript, following points should be considered:

Major

1. Heterogeneity of the datasets

The authors used multiple datasets (i.e., the Common Mind, PsychEncode, GTEx, BrainCloud and ROS-MAP) to build, validate and test their models. Given the potential heterogeneity of the demographic characteristics (e.g. age distribution) of these study cohorts, the authors may need to provide a table comparing the demographic information across these cohorts to better appreciate the variables, which can be presented in a table including but not limited to age, sex, race, and education.

2. Linearity of the data

In the current study, the molecular age was calculated with linear model (i.e. elastic net regression). However, the linearity of the data decreases in older group as indicated by the lower R value in CM-older and PE-older group compared with their younger counterparts (this is also obvious by visual inspection of the scatter plot). This suggests that the linear model may not be the optimal model. The authors may need to re-evaluate the model and to make a comparison between the linear and non-linear models.

3. Interaction between chronological age and molecular age (related to Fig 4 and 5, and Table 1)

a. It would be interesting if the authors could clearly present how these two factors correlate with AD and PD-related clinical traits in both additive and synergistic manners. Specifically, is there any age range where the association of molecular aging with these traits is most pronounced (as is often the case between genetic risk factor and brain disorders)? Or, is the impact of molecular aging uniform across chronological aging?

In addition, it would be helpful if the authors could provide a scatter plot showing the relationship between chronological age and the residuals (i.e. Δ age).

b. Similarly, are there any sex-dependent differences in their interactions?

4. Synergistic interaction between molecular aging and APOE4

a. In Fig 6a, the authors elegantly showed that molecular aging and APOE4 are synergistic risk factors for AD. However, it would be interesting to know whether Δ age has a dose-dependent effect. To show that, the authors can re-bin the data and provide bar chart with more Δ age groups (e.g. -9, -6, -3, 0, +3, +6, +9 years).

b. In Fig 6b, the authors suggested a model to explain the synergistic interaction between Δ age and APOE4. However, as shown in Fig5i, in individuals with one APOE4 allele, the Δ age is only slightly higher (less than 5 years, $p = 0.04$), while in APOE4 homozygous, the Δ age is not significantly different from that of the control (This result is somewhat unexpected as APOE4 shows dose-dependent effect for AD risk). Does this mean that the model proposed by the authors is not in agreement with the data? Please comment on this point.

5. Insight into the APOE4-driven AD pathogenesis

In Fig 6a, the risk increasing effect of APOE4 is not observed in subjects with Δ age of -5 to 0 years. How do their clinico-pathological traits look like? Are they resistance to AD or resilient to AD (i.e., cope with AD pathology)? Please clarify.

6. Overall conclusion

a. It is the reviewer's opinion that the results must be interpreted in a scientifically fair fashion to reflect overall conclusion including the title of the manuscript. In this regard, the authors might want to be more careful in describing as to whether the molecular aging is causally involved in shaping clinical traits. For example, what if APOE4 just drives AD pathology leading to the enhanced signatures of molecular aging (as a consequence), leading to the results in Fig 6a?

b. Related to #6a, what is the similarity and difference between molecular aging signatures (age-sensitive transcripts) and AD-related signatures?

Minor

1. In page 4, "...old to nearly 40% in people over 90 years old" should be "...old to nearly 40% in people over 90 years old in the US" (according to the reference)

2. In page 10-11, the description "It may be observed that most transcripts show continuous incremental differences..." may not be accurate. In Fig 4a. PsychEncode Cohort shows gradual change of gene expression for the increasing genes, while for decreasing genes, the expression seems to be more uniform; In common Mind cohort. The heatmap shows a more discrete rather than a continuous pattern. Please also add a color bar in this figure.

3. In page 12, "Interestingly, another three of the...", should be "Interestingly, the other three categories of..."

4. In page 16, "While most brain-related diagnoses and phenotypes associated significantly with Δ age...", should be "... are associated..."

5. In page 20, "Other top categories of transcripts reduced in the aging brain...and other synaptic functions" should be "... synaptic function-related proteins"

6. Figure 6b is not mentioned in the main text.

Reviewer #1 (Comments to the Authors (Required)): Summary:

The manuscript "Rate of brain aging and APOEε4 are synergistic risk factors for Alzheimer's disease" by Glorioso et al describes (1) the development of a measure of molecular brain age using transcriptome data from the postmortem brains of healthy human subjects aged 25-89 (N=239), and (2) the application of this measure to transcriptome data from the postmortem brains of 4 additional cohorts of healthy human subjects and a large group (N=438) of subjects with and without brain

correlated with chronological age and DNA methylation age. The authors found that the group of subjects with older molecular brain ages (relative to chronological age) were enriched in subjects with Alzheimer's disease, Parkinson's disease, and cognitive decline. In contrast, the authors found that the group of subjects with younger molecular brain ages were depleted in subjects with Alzheimer's disease, even among those subjects with the APOEε4 allele.

Major Comments:

This study addresses an important question re: the relationship among age-related transcriptome changes, late-life brain disease/dysfunction, and APOEε4. The use of data from a large number of subjects from multiple brain tissue collections improves the generalizability of the findings. Integrating the analyses of molecular brain aging with that of APOE genotype adds value. The conclusions are well-supported but weaknesses exist that limit the strength of the conclusions that can be drawn.

In my view, the major weakness of the manuscript is the limited and non-uniform description of the cohorts. For each cohort, the approach to arriving at a diagnosis, or absence of a diagnosis, were different but this is not easy to discern from the methods section. The descriptions of the cohorts all contain different information, for example, RIN value (standard deviation) is reported for the Common Mind cohort but not for any of the others. The manuscript would benefit from a single table with the following values for each cohort: mean age, mean PMI, mean RIN, number of male and female subjects, number of subjects of each race. If this data is not available, then that should be explained.

We have now created this table and added it to the main text (Table 1).

Further, since the measure of molecular brain aging was developed using 239 subjects from the Common Mind cohort, the supplemental information should include subject-level information for age, PMI, RIN, sex, race, tissue bank.

We have created a table of subject level characteristics (Supplemental Table 1)

Also for the supplemental information, delta age versus PMI and RIN for the 239 subjects from the Common Mind cohort should be plotted.

We included all potentially confounding variables such as PMI and RIN in our linear models at all steps so these should not influence our results.

Additional Comments:

Was there any evidence suggesting that APOEε2 allele was associated with younger molecular brain ages?

There was not. The p-value of the regression of APOE2 and delta age is p=0.4.

Neuron cell death is repeatedly referred to as being a part of normal brain aging (p.7, p.11, p.21). It is my understanding that neuron cell death and normal brain aging are mutually exclusive phenomena. What is meant in these passages needs to be clarified.

We agree that there is no cell death during normal aging in the Prefrontal cortex, the area of the brain that we are investigating in this manuscript. Neuron cell death does occur during normal aging in certain areas of the brain such as the Striatum. We emphasize that there is no neuron loss because it is a frequent question that we receive due to confusion with other brain areas and older papers with confounds that other scientists are familiar with. We believe it strengthens the paper to also address this question scientifically, showing that we see no cell death during normal aging in our cohorts as expected.

What gene list was used as background for the Ingenuity analyses?

Ingenuity uses a built in background. Details of their statistical analysis can be found here:

<http://qiagen.force.com/KnowledgeBase/KnowledgeIPAPage#> . We further checked our results with the negative control of analyzing transcripts that did not change with age p>0.5 as a separate analysis. The top categories that we found for transcripts that are age-regulated were not represented when using this negative control list.

Pathway analyses were performed for transcripts that increase with age and transcripts that decrease with age but not for all transcripts that change with age. What is the rationale for this?

We performed pathway analysis on both the set of all age-sensitive transcripts and the set split into increasing and decreasing with age. We obtained similar results either way. We felt that it was more informative to present the data split into increasing and decreasing to provide more context. For example, it seems useful to know that transcripts associated with the category of immune cell trafficking are increasing with age while those related to Sirtuin signaling are decreasing with age.

Reviewer #2 (Comments to the Authors (Required)):

In this manuscript, Glorioso et al. developed a computational model to determine the molecular age (but not chronological age) of the individual brain transcriptome data, using transcriptomic datasets from a cognitively healthy cohort. After a validation using methylation dataset, the authors applied this model to investigate the relationship between molecular brain aging and clinical traits using ROS-MAP data. They found that advanced molecular aging of the brain is associated with major AD and PD-related phenotypes.

Furthermore, they built a model in which interactions between APOE4, molecular aging and AD-related phenotypes were incorporated, and concluded that molecular aging and APOE4 are synergistic risk factors for AD. Linking molecular basis of

aging with that of brain disorders is not conceptually novel; however, the use of multiple transcriptomic datasets and methylation data to define molecular aging signatures supports the rigor of the study. In addition, modeling between APOE4, molecular aging and AD-related phenotypes is of interest, as it might provide insight into the APOE4-driven AD-related pathogenesis. Overall, this is a well-written and interesting manuscript. To further strengthen this manuscript, following points should be considered:

Major

1. Heterogeneity of the datasets

The authors used multiple datasets (i.e., the Common Mind, PsychEncode, GTEx, BrainCloud and ROS-MAP) to build, validate and test their models. Given the potential heterogeneity of the demographic characteristics (e.g. age distribution) of these study cohorts, the authors may need to provide a table comparing the demographic information across these cohorts to better appreciate the variables, which can be presented in a table including but not limited to age, sex, race, and education.

We have now created this table and added it to the main text (Table 1).

2. Linearity of the data

In the current study, the molecular age was calculated with linear model (i.e. elastic net regression). However, the linearity of the data decreases in older group as indicated by the lower R value in CM-older and PE-older group compared with their younger counterparts (this is also obvious by visual inspection of the scatter plot). This suggests that the linear model may not be the optimal model. The authors may need to re-evaluate the model and to make a comparison between the linear and non-linear models.

We initially fit exponential, logarithmic, and linear models to the molecular age model. They performed almost equally well and make a negligible difference in the delta ages calculated. We chose the simplest fit, linear, in order to avoid overfitting the data as the biggest differences are in the tails(very youngest and oldest subjects) for which we have the least subjects. These fits are shown below. The linear trendline is grey, the logarithmic is orange, and the exponential is blue. R-values were determined by Pearson correlation.

3. Interaction between chronological age and molecular age (related to Fig 4 and 5, and Table 1)

a. It would be interesting if the authors could clearly present how these two factors correlate with AD and PD-related clinical traits in both additive and synergistic manners. Specifically, is there any age range where the association of molecular aging with these traits is most pronounced (as is often the case between genetic risk factor and brain disorders)? Or, is the impact of molecular aging uniform across chronological aging?

We only have AD and PD related traits in the ROS-MAP cohort, which has a narrow and very old age range (avg. age 89 years). We therefore are unable to perform this analysis.

In addition, it would be helpful if the authors could provide a scatter plot showing the relationship between chronological age and the residuals (i.e. Δ age).

There is a significant inverse relationship between delta age and chronological age as shown below in the leftmost panel. We have now included this figure in the supplemental information. This is driven by older subjects as shown in the middle and rightmost panels, as younger subjects (<60 years of age) do not show a relationship of delta age and chronological age whereas subjects older than 60 years do show this relationship. We believe that this may be due to a survivor effect; ie. the oldest subjects have brains that appear younger because those subjects are successful agers. This is consistent with the inverse relationship between delta age and chronological age that has also been shown previously using methylation-based biological age in blood (Genome Biol. 2015 Jan 30;16:25. doi: 10.1186/s13059-015-0584-6). DNA methylation age of blood predicts all-cause mortality in later life.) We include chronological age in our regression models in ROS-MAP to control for this effect and the effect of chronological age on AD and other variables.

b. Similarly, are there any sex-dependent differences in their interactions?

Yes, as might have been expected based on differences in lifespan, men have significantly older delta ages compared to women (shown below and now included in the supplemental info). We control for sex in our regression models.

4. Synergistic interaction between molecular aging and APOE4

- a. In Fig 6a, the authors elegantly showed that molecular aging and APOE4 are synergistic risk factors for AD. However, it would be interesting to know whether Δ age has a dose-dependent effect. To show that, the authors can re-bin the data and provide bar chart with more Δ age groups (e.g. -9, -6, -3, 0, +3, +6, +9 years).

We attempted this analysis and unfortunately the extremes of the delta ages have too few subjects ($n < 3$) in the categories (-9, -6, +6, +9) to create meaningful odds ratios.

- b. In Fig 6b, the authors suggested a model to explain the synergistic interaction between Δ age and APOE4. However, as shown in Fig 5i, in individuals with one APOE4 allele, the Δ age is only slightly higher (less than 5 years, $p = 0.04$), while in APOE4 homozygous, the Δ age is not significantly different from that of the control (This result is somewhat unexpected as APOE4 shows dose-dependent effect for AD risk). Does this mean that the model proposed by the authors is not in agreement with the data? Please comment on this point.

Because there are only four homozygous APOE4 subjects in the cohort, we are not powered to perform a separate analysis comparing these subjects to those without APOE4. This is why we instead performed logistic regression including both heterozygous and homozygous APOE4 subjects.

5. Insight into the APOE4-driven AD pathogenesis

In Fig 6a, the risk increasing effect of APOE4 is not observed in subjects with Δ age of -5 to 0 years. How do their clinico-pathological traits look like? Are they resistance to AD or resilient to AD (i.e., cope with AD pathology)? Please clarify.

There is a non-significant difference in the amount of amyloid in young (-5 to 0) delta age subjects and other subjects ($p=0.71$) and the amount of tangles between (-5 to 0) delta age subjects and all other subjects ($p=0.84$) using a one-tailed Student's t-test. Using linear regression, we see a barely significant ($p=0.04$) relationship of delata age to tangles but not to amyloid (Table 2 main text). It therefore appears that delta age is not related to risk of AD and cognitive aging solely by affecting AD pathology, which is in contrast to APOE4's highly significant relationship to pathology. We think this is an interesting point and now mention it in the manuscript text.

6. Overall conclusion

- a. It is the reviewer's opinion that the results must be interpreted in a scientifically fair fashion to reflect overall conclusion including the title of the manuscript. In this regard, the authors might want to be more careful in describing as to whether the molecular aging is causally involved in shaping clinical traits. For example, what if APOE4 just drives AD pathology leading to the enhanced signatures of molecular aging (as a consequence), leading to the results in Fig 6a?

We agree that it is difficult to completely untangle cause and effect with delta age, APOE4, and pathology without a longitudinal living study with a proxy for brain aging. The fact that delta age has a much less significant relationship to AD pathology than APOE4 (discussed above and shown by regression in table 2) suggests that it likely is working through a somewhat different mechanism. Additionally, in support of this, delta age relates to cognitive aging in subjects without neurological disease. We attempted to convey the nuance of this in the discussion and are open to reviewer suggestions on wording.

- b. Related to #6a, what is the similarity and difference between molecular aging signatures (age-sensitive transcripts) and AD-related signatures?

A much larger number of transcripts are significantly related to aging (n=1294) than AD (n=86). A vennn diagram of the number of changes at and adjusted pvalue of 0.001 (cutoff used to create the molecular age model) are shown below. There is a significant positive correlation between the changes.

Minor

1. In page 4, "...old to nearly 40% in people over 90 years old" should be "...old to nearly 40% in people over 90 years old in the US" (according to the reference)

We have now added this.

2. In page 10-11, the description "It may be observed that most transcripts show continuous incremental differences..." may not be accurate. In Fig 4a. PsychEncode Cohort shows gradual change of gene expression for the increasing genes, while for decreasing genes, the expression seems to be more uniform; In common Mind cohort. The heatmap shows a more discrete rather than a continuous pattern. Please also add a color bar in this figure.

Plots of data points for individual transcripts show continuous incremental changes. We have now added a color bar.

3. In page 12, "Interestingly, another three of the...", should be "Interestingly, the other three categories of..."

We have now corrected this.

4. In page 16, "While most brain-related diagnoses and phenotypes associated significantly with Δ Age...", should be "... are associated..."

We have now corrected this.

5. In page 20, "Other top categories of transcripts reduced in the aging brain...and other synaptic functions" should be "... synaptic function-related proteins"

We have now corrected this.

6. Figure 6b is not mentioned in the main text.

We have now corrected this.

Sincerely,

Leonard Guarente, PhD
Novartis Professor of Biology
Massachusetts Institute of Technology

April 18, 2019

RE: Life Science Alliance Manuscript #LSA-2019-00303R

Dr. Leonard Guarente
MIT
Dept. of Biology
77 Massachusetts Avenue
Cambridge, USA-Cambridge, MA 02139-4307 02139

Dear Dr. Guarente,

Thank you for submitting your revised manuscript entitled "Rate of brain aging and APOE ϵ 4 are synergistic risk factors for Alzheimer's disease". As you will see, the reviewers appreciate the introduced changes and we would thus be happy to publish your paper in Life Science Alliance pending final revisions necessary to meet our formatting guidelines:

- please link your profile in our submission system to your ORCID iD, you should have received an email with instructions on how to do so. Please ask the second corr author to do the same
- please add callouts to all figure panels for figure 2 (currently only A is mentioned in the text)

A. FINAL FILES:

B. MANUSCRIPT ORGANIZATION AND FORMATTING:

Sincerely,

Reviewer #1 (Comments to the Authors (Required)):

The authors adequately addressed my concerns.

Reviewer #2 (Comments to the Authors (Required)):

The authors have adequately addressed reviewers' comments. I have no further concerns.

May 10, 2019

RE: Life Science Alliance Manuscript #LSA-2019-00303RR

Dr. Leonard Guarente
MIT
Dept. of Biology
MIT
77 Massachusetts Avenue
Cambridge, USA-Cambridge, MA 02139-4307 02139

Dear Dr. Guarente,

Thank you for submitting your Research Article entitled "Rate of brain aging and APOE ϵ 4 are synergistic risk factors for Alzheimer's disease". It is a pleasure to let you know that your manuscript is now accepted for publication in Life Science Alliance. Congratulations on this interesting work.

DISTRIBUTION OF MATERIALS:

Again, congratulations on a very nice paper. I hope you found the review process to be constructive and are pleased with how the manuscript was handled editorially. We look forward to future exciting submissions from your lab.

Sincerely,
